# Analysis of CD74 Occurrence in Oncogenic Fusion Proteins

**DOI:** 10.3390/ijms242115981

**Published:** 2023-11-05

**Authors:** Jasmine Vargas, Georgios Pantouris

**Affiliations:** Department of Chemistry, University of the Pacific, Stockton, CA 95211, USA; j_vargas16@u.pacific.edu

**Keywords:** cluster of differentiation 74 (CD74), oncogenic fusion protein, fusion gene, CD74-ROS1, CD74-NTRK1, CD74-NRG1, CD74-NRG2α, CD74-PDGFRB, cancer

## Abstract

CD74 is a type II cell surface receptor found to be highly expressed in several hematological and solid cancers, due to its ability to activate pathways associated with tumor cell survival and proliferation. Over the past 16 years, CD74 has emerged as a commonly detected fusion partner in multiple oncogenic fusion proteins. Studies have found CD74 fusion proteins in a range of cancers, including lung adenocarcinoma, inflammatory breast cancer, and pediatric acute lymphoblastic leukemia. To date, there are five known CD74 fusion proteins, CD74-ROS1, CD74-NTRK1, CD74-NRG1, CD74-NRG2α, and CD74-PDGFRB, with a total of 16 different variants, each with unique genetic signatures. Importantly, the occurrence of CD74 in the formation of fusion proteins has not been well explored despite the fact that ROS1 and NRG1 families utilize CD74 as the primary partner for the formation of oncogenic fusions. Fusion proteins known to be oncogenic drivers, including those of CD74, are typically detected and targeted after standard chemotherapeutic plans fail and the disease relapses. The analysis reported herein provides insights into the early intervention of CD74 fusions and highlights the need for improved routine assessment methods so that targeted therapies can be applied while they are most effective.

## 1. Introduction

A fusion protein is the outcome of a multistep process occurring at either the DNA or RNA level and involves the cleavage of two independent genes, followed by their joining into a hybrid one. Upon translation, the newly formed fusion protein may or may not have a biological activity depending on multiple factors, such as the protein’s localization, structural architecture, and functional domains [1]. Fusion proteins occurring in healthy individuals are of low significance as they are often inactive. However, in cancer, the identification of such proteins, which are widely known as oncogenic fusion proteins, has great diagnostic and therapeutic value [2]. Notably, oncogenic fusion proteins are cancer-specific, bioactive molecules that have been detected in both hematological and solid cancers [3]. The identification of fusion proteins is challenging and primarily performed by DNA or RNA next-generation sequencing (NGS), fluorescence in situ hybridization (FISH), reverse transcription-polymerase chain reaction (RT-PCR), and reverse transcription-quantitative polymerase chain reaction (RT-qPCR). Via these methods, which may be used alone or in combination, a large number of oncogenic fusions have been detected in either cancer patients or retrospective studies. Several oncogenic fusions protein families exist, with ROS proto-oncogene 1 (ROS1), anaplastic lymphoma kinase (ALK), neurotrophic tyrosine receptor kinase (NTRK), and neuregulin 1 (NRG1) being some of the most common partners that are primarily detected in lung cancers [4,5,6].

Cluster of Differentiation 74 (CD74) is a membrane-bound protein expressed on the surface of human antigen-presenting cells (APCs). Also known as the invariant chain (Ii), the soluble form of CD74, which is found in the endoplasmic reticulum (ER), associates with major histocompatibility class II (MHC II) molecules to ensure their proper folding. CD74 can act as a receptor for the tumorigenic and proinflammatory cytokine macrophage migration inhibitor factor (MIF) [7] and D-dopachrome tautomerase (D-DT, or MIF-2) [8]. After the formation of either the MIF-CD74 or D-DT-CD74 complex, intracellular signal transduction is achieved through the presence of the coreceptor protein, CD44 [9]. Other receptors capable of forming a complex with CD74 for MIF signaling include CXCR2 [10] and CXCR4 [11]. Consequently, the extracellular signal-regulated kinase (ERK)-1/2 mitogen-activated protein kinase (MAPK) [8,12], nuclear factor kappa B (NF-κB) [13], phosphoinositide-3-kinase (PI3K)/Akt [14] and c-Jun N-terminal kinase (JNK) pathways are activated [15] with downstream effects in cell survival and proliferation [16].

Besides its expression on immune system cells, CD74 has also been expressed in a variety of cancer cells, including thyroid carcinoma [17], bladder cancer [18], chronic lymphocytic leukemia (CLL) [19], multiple myeloma (MM) [20], breast cancer [21], gastrointestinal cancers [22], non-small cell lung cancer (NSCLC) [23], renal cell carcinoma (RCC) [24], prostate cancer [25], pancreatic cancer [26], and glioblastomas [27]. The elevated expression of CD74 can be a marker of tumor progression [28] as well as poor clinical prognosis [29]. Since the beginning of the 21st century, CD74 has also been reported to participate in the formation of oncogenic fusion proteins [30,31,32,33,34]. While the functional role of CD74 in these oncogenic fusion proteins is completely unknown, in certain protein families, such as ROS1 and NRG1, its occurrence is very high.

Transcriptional splicing can produce four different human CD74 isoforms, p33, p35, p41, and p43, which are 216 [35], 232 [36], 280 [37], and 296 [38] amino acids long, respectively [37] (Figure 1 and Appendix A). Structurally, CD74 is encoded by 8–9 exons expressing an N-terminal intracellular cytoplasmic tail (exon 1), a single transmembrane segment (exon 2), and an extracellular C-terminal domain (exons 3–8) [39], which includes exon 6, a critical factor for trimerization [40]. The ninth exon arises from alternative splicing that produces exon 6b, which encodes in both the longer CD74 isoforms, p41 and p43 [39]. Similarly, an alternative start codon gives rise to the 16 amino acid (aa) N-terminal extension, which is present in the p35 and p43 CD74 isoforms [41]. The biological assembly of CD74 is a homotrimer [42] and characterized by enhanced flexibility that has negatively impacted all efforts towards obtaining complete structural information. Instead, structural fragments of CD74 have been resolved by nuclear magnetic resonance (NMR) spectroscopy, or protein crystallography and deposited in the protein data bank (PDB). These include the trimeric domain [43], class II-associated invariant chain peptide (CLIP) [44,45,46,47], and the p41 fragment [48,49] (Figure 1).

Drugs that target CD74 are limited despite its great clinical interest. As of 2008, the anti-CD74 monoclonal antibody (mAb), milatuzumab, received the orphan drug designation in the United States (U.S.) for the treatment of MM. Phase 1 clinical trials in MM and B-cell lymphoma patients found that Milatuzumab did not achieve an objective response rate (ORR), but patients were able to achieve stable disease with treatment [52,53]. Ten years later, in 2018, the anti-CD74 antibody drug conjugate (ADC), STRO-001, also received the same designation. Phase 1 clinical trials of STRO-001 in MM and B-cell lymphoma patients are currently ongoing [54]. Despite these efforts, to date, there is still no recognized Food and Drug Administration (FDA)-approved drug for the treatment of CD74 overexpression. While there are antibodies available to target this receptor, a small-molecule antagonist has yet to be identified. As CD74 is implicated in a diverse range of cancers, it is important to discover novel and effective ways to meet the challenges associated with this protein.

This review is focused on the CD74 oncogenic fusions (CD74-ROS1, CD74-NTRK1, CD74-NRG1, CD74-NRG2α, and CD74-PDGFRB) identified in various cancers. The structural features and expression patterns of all known CD74 fusion variants are described. Clinical patient characteristics, treatment plans, detection methods, as well as retrospective analyses of tumor samples are also included to provide insights into CD74 related oncogenic fusion proteins. While the CD74 protein alone is an attractive molecular target in multiple cancers, the study of cancer-specific proteins, such as the ones described here, offer a great opportunity for refining currently utilized cancer treatment plans and developing new methods for early detection.

## 2. CD74 Fusion Proteins in Cancer

To date, five in-frame CD74 fusion proteins have been reported in cancer. These include CD74-NRG2α, CD74-NTRK1, CD74-ROS1, CD74-PDGFRB, and CD74-NRG1 (Figure 2). The discovery of CD74 fusion proteins began with the identification of CD74-ROS1 in an adult female lung adenocarcinoma patient in 2007 [30] (Figure 3). This fusion protein is characterized by the presence of two transmembrane domains: one obtained from CD74 and the other from ROS1. According to the structural architecture of this fusion, the kinase domain of ROS1 is localized in the cytoplasm (Figure 2). In 2013, CD74-NTRK1 was the next reported fusion, which was detected in an adult patient (unspecified gender) with lung adenocarcinoma (Figure 3) [31]. Similar to CD74-ROS1, this oncogenic fusion protein contains two transmembrane domains and an intracellular kinase domain [31] (Figure 2). An adult female invasive mucinous adenocarcinoma patient was found to harbor a CD74-NRG1 fusion in 2014 [32] (Figure 3). In contrast to the first two reported CD74 fusion proteins CD74-NRG1 has only one transmembrane domain, which is derived from CD74 [32]. Upon the formation of the CD74-NRG1 fusion, the epidermal growth factor (EGF)-like domain of NRG1 remains intact in the extracellular space, providing the ligand for ERBB2-ERBB3 receptor complexes. In 2016, a study focused on adult and pediatric patients with B-cell acute lymphoblastic leukemia (B-ALL) revealed 29 in-frame fusion proteins, one of which was the novel CD74-PDGFRB (Figure 3). This fusion was identified in a pediatric patient, and according to its occurrence, is significantly rarer in comparison to CD74-ROS1, CD74-NTRK1, and CD74-NRG1 [33]. From the structural point of view, CD74-PDGFRB contains two transmembrane domains, one from each protein partner and a split kinase domain found in the cytoplasm (Figure 2). In 2020, CD74-NRG2α was identified in a female acinar adenocarcinoma patient from Japan [34] (Figure 3). This protein is the latest known CD74 fusion, and similar to CD74-PDGFRB, is very rare. CD74-NRG2α contains one transmembrane domain, from CD74, and an extracellular EGF motif encoded by exons 4 and 5 of NRG2α (Figure 2).

Noteworthily, the majority of CD74 fusion proteins found in patients were first reported in lung cancers, and the same trend continues to date (Figure 3). Of the five fusions, only CD74-ROS1 was additionally identified in an inflammatory breast cancer patient (Appendix A) [55], besides its presentation in lung cancer. Interestingly, the rare CD74-PDGFRB oncogenic fusion remains relevant to cancer in exclusively leukemia [56] (Appendix A).

CD74-ROS1 has seven variants reported in the literature, while CD74-NTRK1 and CD74-NRG1 have three and four variants each, respectively (Table 1). CD74-PDGFRB and CD74-NRG2α are very rare, with only one known functional fusion (Table 1). Throughout most of the fusion variants, the first six exons of CD74, which include the intracellular, transmembrane, and extracellular trimeric domain, are retained. Of the several isoforms of oncogenic fusion proteins identified, only the amino acid sequences of CD74-ROS1 (C6-R34 [30,57] and C6-R32 [57]) and CD74-NTRK1 (C8-N12 [31]) are published (Appendix A). The total amino acid residues reported for CD74-ROS1 variants C6-R34, C6-R32, and C6-R33 are 703 [30,57], 806 [57], and 778 aa [58], respectively. Of the CD74-NTRK1 variants, the C8-N12 one has been documented as 627 aa long [31]; CD74-NRG1 C6-N6 was reported as 283 aa [32] and 282 aa [59] long in two separate publications. Of the remaining fusions, CD74-NRG1 C8-N6 is documented as 303 aa in length [59]. Of note, the amino acid length of CD74-PDGFRB and CD74-NRG2α have not been reported.

### 2.1. Analysis of CD74 Fusion Partners

In an expression study of human cancer cell lines, ROS1 was found highly expressed in many glioblastoma cell lines but was not seen in normal healthy brain tissue [68]. ROS1 has also been found to be overexpressed in oral carcinoma [69] and lung cancer tissue [70,71]. From the cell line SW-1088, *ROS1* was further characterized to encode a 259 kDa (2347 aa) protein with a glycosylated extracellular domain, a single transmembrane domain, and an intracellular tyrosine kinase domain [72]. Despite the early characterization, the ROS1 ligand is still unknown, making the protein an orphan receptor [73]. The *ROS1* gene is encoded by 43 exons according to the NCBI Reference Sequence NM_002944.2. Exons 1–32 produce the extracellular domain of ROS1, while exons 33 and exons 36–43 encode the transmembrane portion and C-terminal domain, respectively. The intracellular kinase domain or ROS1 is encoded by exons 36–41 [74]. Tumors positive for ROS1 are typically treated with crizotinib, a tyrosine kinase receptor (TRK) inhibitor. This treatment plan was approved for clinical use in 2016 [75] based on a phase I clinical trial with ROS1 rearrangements in NSCLC that had a 72% ORR to the drug [76]. In 2019, entrectinib, a pan-TRK inhibitor, was approved for use against ROS1–positive NSCLC fusions based on its 77% ORR in clinical trials and ability to penetrate the central nervous system, making it suitable for ROS1-fusion NSCLC patients who experienced brain metastasis [77].

The *NTRK1* gene encodes the tropomyosin kinase receptor A (TRKA) protein, which is involved in neural development [78]. This protein is produced in two isoforms due to alternative splicing [79]: one is 790 aa long (trkAI) [79,80] and the more abundant form is 796 aa (trkAII) [79,81]. TRKA is composed of an extracellular domain for ligand binding, a transmembrane domain, and an intracellular tyrosine kinase domain [80]. Nerve growth factor (NGF) is the preferred ligand that binds with high affinity to the TRKA receptor [82,83]. The main signaling pathways activated by TRKA include PI3K-AKT, Phospholipase C (PLC)-γ, ERK, and the MAPK pathways, which are all needed for neuronal survival and differentiation [84,85]. Human TRKAII, encoded by the *NTRK1* gene, contains 17 exons; exons 1–9 encode a unique extracellular domain; exons 10–11 yield the transmembrane domain; and exons 11–17 produce for the intracellular domain, which includes the tyrosine kinase domain [86]. In response to the NGF, the tyrosine kinase domain becomes phosphorylated, which is imperative for intracellular signaling [86]. TRKA was identified as an oncogenic driver when it was found mutated in an acute myeloid leukemia (AML) patient [87]. The findings of the case study showed that the deletion of a 75-amino acid segment from the extracellular domain led to the constitutive tyrosine phosphorylation of the protein. Besides AML, TRKA has also been reported to participate in multiple fusions proteins [81]. The overexpression of TRKA is associated with many other cancers, including lung [88], thyroid [89], breast [90], and cervical cancer [91]. Oncogenic fusions typically incorporate the C-terminal portion of NTRK that contains the kinase domain with the N-terminal of a new protein partner. Consequently, the ligand binding domain of NTRK is removed [75]. Larotrectinib, a TRK inhibitor, has been approved for use in patients harboring NTRK rearrangements based on its efficacy in clinical trials [92,93] with a 79% ORR [94]. Entrectinib is also approved for use on NTRK fusions based on its anti-tumor activity and 57% ORR in clinical trials [95].

The *NRG1* gene is known to produce many isoforms, which are expressed across various tissues through different promoters and alternative splicing [96,97]. Furthermore, the expression of NRG1 is important to establish the cardiovascular and nervous systems during embryonic development [98,99]. ERBB3 and ERBB4 are the EGF protein receptors for the NRG1 and NRG2 ligands [100]. This ligand-binding induces receptor heterodimerization with ERBB2, causing the activation of the PI3K and MAPK pathways, which results in increased cell proliferation/migration and resistance to apoptosis in cancer [100,101,102,103]. While the normal biological function of NRG2 remains to be fully understood [104], the structures of NRG1 and NRG2 both have an extracellular EGF-like domain, transmembrane domain, and cytosolic tail [97,105]. While there is currently no FDA-approved treatment for NRG1 fusions, seribantumab, an anti-ERBB3 (HER3) mAb [106], has been used in clinical trials for patients with NRG1 fusion-positive solid tumors with a 30% ORR [107,108]. Likewise, zenocutuzumab, an ERBB2/ERBB3 (HER2/HER3) bispecific antibody, is under evaluation in clinical trials for patients with NRG1-positive fusions in cancer and has a 34% ORR [109].

The platelet-derived growth factor receptor β (PDGFRB) protein is encoded by the *PDGFRB* gene and is known to express in mesenchymal [110] and smooth muscle cells [111]. This receptor tyrosine kinase (RTK) is made up of an extracellular ligand binding domain, a transmembrane domain, and a split tyrosine kinase domain within its intracellular portion [112]. PDGFRB is 962 aa long, where exon 11 precedes the transmembrane domain, exon 12 encodes the juxtamembrane domain, exon 14 encodes the first kinase domain, and exon 18 encodes for the second kinase domain [113]. Once bound to its ligand, PDGF-BB [114], the Ras-MAPK [115] signaling pathway, is activated [112]. PDGFRB was found to be overexpressed in various cancers, including lung [116], colon [117], melanoma [118], and breast carcinoma [119]. Studies found that imatinib, a TKI, achieved durable response in patients with PDGFRB fusions implicated in chronic myeloproliferative disorders [120,121]. Besides imatinib, crenolanib is another TKI that is capable of inhibiting PDGFRB and has shown promising results in lung [122] cancer and colon [123] cancer cell lines.

### 2.2. CD74-ROS1 Expression

The first CD74 fusion protein, CD74-ROS1, was originally identified in NSCLC [30] and 14 years later was also reported in breast cancer [55]. This fusion appears in adult patients [30] and has a membrane-bound cellular localization [30,124] with reports in the plasma membrane [124] and the endoplasmic reticulum [125]. Of the five known CD74 fusions, CD74-ROS1 is the most prevalent among ROS1 fusion proteins in NSCLC [74,126,127,128], and based on the data collected in this review, also appears to be the most prevalent partner among CD74-related fusions. CD74-ROS1 exists in seven different variants (Table 1) that are generated from rearrangements in both the *CD74* and *ROS1* genes. Of the CD74 isoforms, the p35 variant is involved in these fusions, and in patients has been found spliced at different exon breakpoints. Likewise, the *ROS1* rearrangement occurs at different exons, resulting in the formation of unique CD74-ROS1 fusion proteins. CD74-ROS1 fusions typically have breakpoints located on ROS1 exon 32 or 34, retaining the ROS1 kinase domain [60] allowing for the use of tyrosine kinase inhibitors (TKIs).

In 2007, the identification of CD74-ROS1 originally occurred in a 50-year-old female non-smoker with a stage IB lung adenocarcinoma [30]. The patient did not show signs of metastasis in the resected 2 cm tumor, but DNA sequencing revealed the presence of a CD74-ROS1 fusion, exhibiting the splicing of CD74 exon 6 to ROS1 exon 34. Interestingly, this fusion variant contained two transmembrane domains (Figure 2), and the nucleotide sequence was deposited into GenBank (EU236945) [30]. The C6-R34 variant has been detected by RT-PCR in additional stages of lung adenocarcinoma, including IIA, IIIB, and IV [129]. The next identified CD74-ROS1 fusion breakpoint was a splicing between CD74 exon 6 and ROS1 exon 32 that was published in 2012 [57]. Through FISH analysis and independent confirmation by RT-PCR, this C6-R32 fusion was detected in a 9 mm diameter tumor removed from a 79-year-old female stage IA lung cancer patient [57]. Shortly thereafter, this breakpoint was also identified in a 44-year-old female never-smoker with stage IV lung adenocarcinoma using RT-PCR [130]. A case report published in 2014 revealed a CD74-ROS1 variant of CD74 exon 7 fused to ROS1 exon 32 in a patient with three ROS1 fusion variants identified through RT-PCR [60]. Diagnosed with stage IV lung adenocarcinoma, this patient was a 61-year-old woman with no previous smoking history whose computerized tomography (CT) scan revealed a 3 cm mass and magnetic resonance imaging (MRI) additionally showed evidence of brain metastases [60]. Of note, the DNA sequence for this C7-R32 fusion was not obtained [60]. The case of a CD74-ROS1 fusion in a 64-year-old inflammatory breast cancer patient (IBC) was found through NGS as a combination of CD74 exon 7 and ROS1 exon 34, which may have contributed to rapid disease progression as they died 18 months after tumor discovery [55].

In 2021, the most recently identified CD74-ROS1 variant was reported with a breakpoint fusing CD74 exon 3 to ROS1 exon 34 [61]. This novel fusion was identified through next generation sequencing in a 42-year-old female never-smoker with relapsed stage IVA lung adenocarcinoma [61]. While the patient underwent a partial pulmonary lobectomy after her original diagnosis four years prior, her lung cancer relapsed with lymph node, bone, and brain metastases [61]. Due to the presence of ROS1 tyrosine kinase domain, crizotinib, was used effectively against the tumor. Additional brain metastases developed after crizotinib, which was further treated with entrectinib [61].

The expression of CD74-ROS1 has confirmed oncogenic transformation capacity towards fibroblast cells through the activation of MAPK, SHP-2, and STAT-3 pathways [124]. Additionally, based on the oncoprotein’s expression in noninvasive NSCLC cell lines (H1755, H2009, and H1915), CD74-ROS1 induces an invasive mechanism via the phosphorylation of E-Syt1. This mechanism was further validated in a xenograft model to be both invasive and metastatic [124]. Subcellular localization is critical for CD74-ROS1 variants, as those that localize in the ER have a compromised ability to activate the RAS/MAPK pathway [125].

Retrospective studies of tumor samples containing the CD74-ROS1 fusion were recorded to provide information about fusion variant, specimen source, detection method, age, gender, smoking status, and cancer stage (Appendix A). Many specimens are either from formalin-fixed paraffin-embedded (FFPE) tumor tissue or frozen tumor samples. The oncogenic fusion was identified by means of FISH, using probes that detect the ROS1 portion of the fusion, followed by confirmation using RT-PCR. More recent studies verify fusions by NGS, and as recently as 2023, Nanostring has been able to detect fusions in plasma samples [5]. Many of the published cases of CD74-ROS1 are in female never-smoker patients, while the fusion has been detected from stages I-IV, highlighting the necessity of targeted therapy for early intervention. Of the seven different CD74-ROS1 variants, the C6-R34 one appears the most often (Appendix A). The data also suggest that the occurrence of this fusion is equally likely in both smokers and non-smokers (Appendix A).

Clinical and case studies over the years outline additional details of CD74-ROS1 patients, such as the treatment sequence patients received, whether TKI resistance developed, and their overall survival (OS) (Table 2). Based on the cases reported in the literature, many CD74-ROS1 patients benefit from TKIs by having a greater OS and better response to treatment, compared to patients who receive traditional platinum-based chemotherapy, alone (Table 2). Mutations and metastasis can still arise after taking TKIs, such as crizotinib but can sometimes be overcome by introducing another TKI, such as entrectinib, dabrafenib, or cabozantinib (Table 2). A recently published clinical trial revealed that unecritinib (TKI) achieved an 88.9% ORR for patients harboring CD74-ROS1 fusions, with a median progression-free survival (PFS) of 21.2 months [131]. These results suggested a greater efficacy and lower toxicity compared to the previously used TKIs, including crizotinib and entrectinib [131].

While the clinical and case studies (Table 2) show the occurrence of CD74-ROS1 in many patients with relapsed, advanced stage cancer (III–IV), the retrospective studies conducted on patient tumor samples (Appendix A) reveal that many CD74-ROS1 fusions can be detected at earlier cancer stages (I–II). The data suggest that even though the fourth and even ninth line of treatment can have some success in patients, early detection and intervention to target these fusions may yield improved patient outcomes without the necessity of going through traditional chemotherapies (Table 2).

### 2.3. CD74-NTRK1 Expression

CD74-NTRK1 was first detected by NGS in a lung adenocarcinoma patient tumor sample and confirmed using RT-PCR, followed by FISH [31]. This fusion joined exon 8 of CD74 to exon 12 of NTRK1, thus retaining the NTRK1 kinase domain encoding for the TRKA receptor and was predicted to localize in the plasma membrane [31] (Table 1, Figure 2). The complete cDNA sequence of the fusion was further introduced into Ba/F3 cells, 293T cells, and NIH3T3 fibroblasts, proving that *CD74-NTRK1* was oncogenic through the pERK pathway; the gene fusion was also found to induce tumorigenesis in nude mice [31]. The expression of the CD74-NTRK1 oncoprotein resulted in constitutive TRKA kinase activity, and cell lines harboring the fusion were found to be sensitive to TRKA inhibitors ARRY-470, CEP-701, and crizotinib [31].

Much later, in 2020, next-generation sequencing revealed a unique fusion of CD74 exon 7 with NTRK1 exon 8 (C7-N8) in a 41-year-old female stage IIIB adenocarcinoma patient [64] (Table 1). Additionally, NGS has identified a CD74-NTRK1 C6-N12 fusion mutation with a missense mutation in a lung cancer patient [65] without many further details (Table 1). These retrospective studies on tumor samples are detailed with the diagnosis, variant, method of detection, age, gender, and stage (Appendix A). While not much detailed patient information is available on CD74-NTRK1, the three fusion variants are of interest for future studies as they may share similarities with other CD74 fusion proteins, and thus benefit from similar treatment plans.

### 2.4. CD74-NRG1 Expression

CD74-NRG1 was first identified in a 64-year-old female never-smoker stage IB invasive mucinous adenocarcinoma (IMA) lung cancer patient through transcriptome sequencing [32]. This fusion arose from a somatic genomic event, resulting in a chromosomal rearrangement between CD74 exon 6 and NRG1 III-β3 exon 6 [32]. This novel CD74-NRG1 fusion protein was reported as 283 aa long, retained the transmembrane domain of CD74 and the EGF-like domain of NRG1, resulting in its membrane-bound cellular localization [32]. Another 2014 study published shortly thereafter, detected the C6-N6 fusion pairing in female never-smoker IMA patients aged 68 and 53 at cancer stages IIB and IA, respectively, and reported the fusion as 282 aa long. [59].

H322 and H1568 lung cancer cell lines were virally transduced with the full CD74-NRG1 fusion, as well as with a truncated version of the fusion that lacked the EGF-like domain [32]. It was observed that CD74-NRG1 expressing cell lines had increased levels of p-AKT, p-ERBB2, and p-ERBB3 associated with PI3K/AKT pathway activation compared to cells that do not express the fusion [32]. Additional western blot analysis further supported that phosphorylated AKT and ERBB3 were dependent on the EGF-like domain being present in this fusion [32].

An alternative translocation produced a novel C8-N6 fusion pairing that incorporated the first eight exons of CD74 and the EGF-coding NRG1 exons, resulting in a 303 aa CD74-NRG1 fusion [59]. This fusion was detected in three IMA patients through RNA sequencing and was further validated through the Sanger sequencing of RT-PCR products Patient characteristics included a 55-year-old male 47 pack-year smoker at stage IA cancer, and two female never-smokers, one 78 years old at Stage IA IMA and one 47 years old at stage IB [59]. Additionally, in a 2023 published study, a fusion variant incorporating CD74 exon 6 joined to NRG1 exon 4 was identified in two female NSCLC patients in their 80s [66].

The same study treated EFM-19 reporter cells with media from H1299 lung cancer that expressed CD74-NRG1 and found that the fusion protein caused the phosphorylation of ERBB2 and ERBB3, which suggested the activation of HER2:HER3 autocrine receptor signaling; The phosphorylation of ERK and AKT was also observed [59]. Lapatinib and Afatinib (HER TKIs) were tested on the cells, and as a result, phosphorylation events were suppressed [59]. CD74-NRG1 is capable of activating HER2 and HER3 autocrine signaling through the EGF-like domain of NRG1 in exons 6–7 interacting with HER3, which results in PI3K/AKT signaling and subsequent cell proliferation associated with oncogenic growth [66].

Multiple retrospective studies have been conducted on CD74-NRG1 patient tumor samples (Appendix A). Many patients affected by these fusions were female non-smokers afflicted with either IMA or NSCLC. The clinical CD74-NRG1 patient characteristics of CD74-NRG1 are summarized in Table 3 with treatment courses and lifespan included. This fusion is detected in stages I-IV (Appendix A and Table 3) and treated with platinum-based chemotherapy, monoclonal antibodies, and afatinib (Table 3). Among NRG1 fusions in solid tumors, CD74 is found to be the most prevalent fusion partner [142]. Similar to CD74-ROS1, CD74-NRG1 fusions are capable of being identified as early as stage I cancer (Appendix A). Considering the clinical patient characteristics, this fusion may also benefit from targeted therapy early on, rather than a platinum-based first-line-of-treatment approach (Table 3).

### 2.5. CD74-PDGFRB Expression

RNA seq first identified a CD74-PDGFRB fusion in a 2.4-year-old male with B-cell acute lymphoblastic leukemia. The novel protein was found to be a fusion between CD74 exon 6 and PDGFRB exon 11 [33] (Appendix A). Following two rounds of first-line treatment, the pediatric patient went into complete remission [33] (Appendix A). Recently, another variant of CD74-PDGFRB, spliced between CD74 exon 1 and PDGFRB exon 11, was validated through RNA seq and polymerase chain reaction (PCR) in a 22-month-old female B-ALL patient, although it did not form a fusion protein [56]. Many fusion partners tend to fuse at PDGFRB exon 11 [113,144,145], thus retaining the PDGFRB transmembrane domain and split kinase domain.

Interestingly, in another case, Sanger sequencing showed a retained CD74 intron sequence that coded for a stop codon before the fusion breakpoint [56]. Referred to as CD74^intr^::PDGFRB. This fusion lacked an open reading frame to drive the translation of a functional fusion protein [56]. Additionally, a 2023 published case study identified a PDGFRB::CD74 fusion mRNA in a 19-month-old female patient diagnosed with an aggressive case of cutaneous non-Langerhans cell histiocytosis (NLCH) [146]. Although the biopsy specimen was benign, the mutations that led to the PDGFRB::CD74 fusion mRNA had not been previously associated with pediatric or adult NLCH [146].

While this fusion has been documented in the literature several times, its oncogenic variant remains to be further studied and characterized as the oncogenic signaling pathway that is activated is still unknown. But, it is likely that the presence of CD74-PDGFRB leads to the activation of PDGFRB [146].

### 2.6. CD74-NRG2a Expression

The most recently identified CD74 fusion protein was CD74-NRG2α in a 70-year-old female, never-smoker, stage IIIA acinar adenocarcinoma lung cancer patient [34] (Appendix A). This fusion was identified through RNA seq and confirmed through Sanger sequencing to fuse CD74 exon 6 to NRG2α exon 2, which retained the EGF-like domain of NRG2α and the transmembrane domain of CD74 [34] (Appendix A, Figure 2). While the 43 mm tumor was surgically resected, the tumor returned, and the patient was treated with erlotinib, a tyrosine kinase inhibitor, but its use was discontinued a month later due to skin rash; the patient eventually died of disease 32 weeks later [34] (Appendix A). Immunohistochemistry (IHC) analysis of tumor cells were moderately positive for p-HER4 and negative for p-HER2, p-HER3, and p-EGFR. As the single documented case of CD74-NRG2α, more research is needed to elucidate whether the fusion’s phosphorylation of HER4 alone promotes an oncogenic pathway, and if that is the case, how exactly it occurs [147].

## 3. Conclusions

Methods that are utilized to identify CD74 fusions include DNA or RNA NGS, FISH, RT-PCR, and RT-qPCR. In clinical case studies, these methods typically identify the fusion after cancer recurrence. Both male and females are affected by CD74 fusions, with many female non-smokers reported as carriers.

CD74-ROS1 has the highest number of fusion variants (seven) as well as patient data availability. Upon analysis of the published findings, it becomes clear that CD74-ROS1 is formed from very early to advanced-stage cancers. This fusion is mostly detected in NSCLC and likely occurs equally in both smokers and non-smokers. CD74-NRG1 has the second greatest number of variants (four), followed by CD74-NTRK1 (three), while CD74-NR2α and CD74-PDGFRB have a single fusion protein variant each. CD74-NRG1 is detected often at stages I and II in retrospective studies of patient tumor tissue, while clinical studies report this fusion both in early and late-stage IMA and NSCLC. Based on patient data available, the CD74-NRG1 fusion occurs mostly in non-smokers, with a few cases in smokers.

About half of the CD74 fusion proteins have been studied extensively and their signaling pathways elucidated, mostly in respect to their fusion partner. Likewise, inhibitors target the fusion partner rather than CD74. Throughout the five fusions expressed in cancer patients, all retain the transmembrane domain of CD74 that many times makes up for the transmembrane domain that biologically occurs in the fusion partner before splicing occurs. CD74 fusion proteins mainly manifest in lung cancer, apart from a single documented case in an aggressive case of inflammatory breast cancer and two cases of pediatric leukemia. Of note, the rare fusions CD74-PDGFRB and CD74-NRG2α require further studies to elucidate their exact mechanism of disease.

As many studies focus on the targetable fusion partners, the oncogenic role of CD74 in these CD74 fusion proteins remains to be fully understood. In ROS1 and NRG1 fusion families, CD74 is the most common partner, emphasizing the potential significance of CD74 occurrence in fusion proteins. The participation of CD74 as a partner in five different fusion proteins, with a total of 16 different variants, highlights its involvement in the formation of various types of fusions. The data on CD74 fusions offers insight for improved therapy plans for cancer patients and the development of additional methods for early detection to improve treatment success and patient survival.

## Figures and Tables

**Figure 1 ijms-24-15981-f001:**
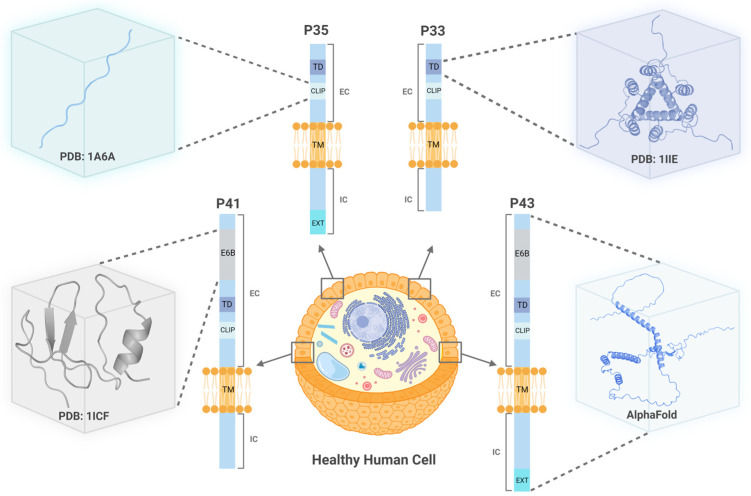
The CD74 isoforms P41 (280 aa—bottom left), P35 (232 aa—top left), P33 (216 aa—top right), and P43 (296 aa—bottom right) depicted on the cell surface of a healthy human cell. Experimentally resolved structural fragments of CD74 are illustrated in boxes along with their corresponding PDB numbers. The predicted structure of P43, produced by AlphaFold Protein Structure Database, is also provided [50,51]; Accension AF-P04233. The size of each domain shown in the figure is drawn with no scale. Light orange color denotes the transmembrane domain. IC: Intracellular, EC: extracellular, CLIP: class II-associated invariant chain peptide, TM: transmembrane domain, TD: trimerization domain, E6B: exon 6b, EXT: N-terminal extension.

**Figure 2 ijms-24-15981-f002:**
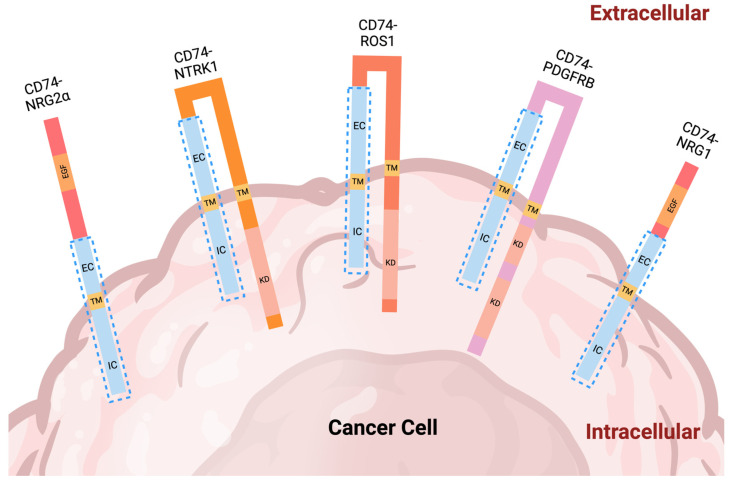
CD74 oncogenic fusions. From left to right, CD74-NRG2α, CD74-NTRK1, CD74-ROS1, CD74-PDGFRB, and CD74-NRG1 are depicted on the membrane of a cancer cell. The CD74 segments participating in the formation of these fusions are enclosed within the dashed rectangles. For each fusion, the functional domain (EGF and KD) known to contribute to the protein’s oncogenic activity is illustrated. In all cases, this domain is derived from the protein partner of CD74. The size of each domain shown in the figure is drawn with no scale. IC: Intracellular, TM: transmembrane domain, EC: extracellular, EGF: epidermal growth factor-like domain, KD: kinase domain.

**Figure 3 ijms-24-15981-f003:**
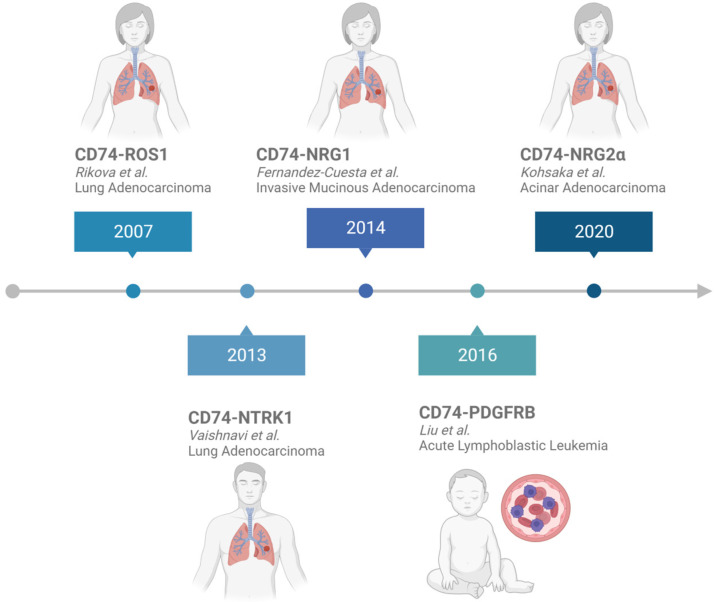
Timeline for the identification of CD74 fusions. The original study, gender of the patient, and type of cancer are also reported. CD74-ROS1 [30], CD74-NTRK1 [31], CD74-NRG1 [32], CD74-NRG2α [33], and CD74-PDGFRB are displayed in chronological order [34]. Of note, the gender of the CD74-NTRK1 patient was not reported in the original study.

**Table 1 ijms-24-15981-t001:** CD74 fusion protein variants detected in cancer patients.

CD74 Fusion Protein	Exon Breakpoint	Cancer	Reference
CD74-ROS1	C6-R34	NSCLC	[30]
	C6-R32	NSCLC	[57]
	C7-R32	NSCLC	[60]
	C7-R34	IBC	[55]
	C3-R34	NSCLC	[61]
	C6-R33	NSCLC	[62]
	C6-R35	NSCLC	[63]
CD74-NTRK1	C8-N12	NSCLC	[31]
	C7-N8	NSCLC	[64]
	C6-N12	NSCLC	[65]
CD74-NRG1	C6-N6	IMA	[32]
	C8-N6	IMA	[59]
	C6-N4	NSCLC	[66]
	C7-N6	IMA	[67]
CD74-PDGFRB	C6-P11	B-ALL	[33]
CD74-NRG2α	C6-N2	NSCLC	[34]

**Table 2 ijms-24-15981-t002:** CD74-ROS1 Clinical Patient Characteristics.

Diagnosis	Variant	Detection Method	Age	Gender	Smoker/Pack Year (PY)	Stage	Treatment	TKI Resistance	Lifespan OS	Reference
NSCLC	C7-R32 C6-R34	RT-PCR	61	F	0	IV (T4N3M1b)	1st: pemetrexed + cisplatin * refused crizotinib for economic reason	-	-	[60]
IBC	C7-R34	NGS	64	F	-	cT4N3M1	1st: paclitaxel 2nd: capecitabine * refused crizotinib for economic reason	-	Died 4 mos. after dx	[55]
NSCLC	C3-R34	NGS	38	F	0	IVA (T1aN3M1A)	1st: cisplatin, pemetrexed, + bevacizumab 2nd: docetaxel + ramucirumab * 3rd: crizotinib * 4th: entrectinib	Crizotinib resistance: brain metastasis	1 mo. PFS	[61]
Metastatic NSCLC	-	CGP (NGS) ctDNA assay	41	F	0	-	1st: carboplatin, pemetrexed + bevacizumab * 2nd: crizotinib	-	PFS 4 mos. after crizotinib	[132]
CBPB	-	NGS	44	F	0	IV (cT3N1M1b)	* 1st: crizotinib	Crizotinib resistance: ROS1 G2032R mutation	PFS for 3 mos. PD after crizotinib resistance	[133]
PPC	-	NGS RT-PCR	56	F	0	IIIA (T2aN2M0)	1st: resection 2nd: paclitaxel, carboplatin + bevacizumab * 3rd: crizotinib	-	CR after 6 mos.	[134]
NSCLC	C6-R34	NGS	40	F	0	-	1st: gemcitabine + cisplatin 2nd: docetaxel 3rd: gefitinib * 4th: cabozantinib * 5th: crizotinib	Cabozantinib resistance	SD after 2 mos. crizotinib	[135]
NSCLC	C6-R33	NGS RT-qPCR	51	F	-	IVB (pT4N3M1c)	1st: pemetrexed + cisplatin * 2nd: crizotinib	-	PFS 15.7 mos. after crizotinib	[58]
Metastatic NSCLC	C6-R33	NGS	60	F	0	-	1st: radiotherapy * 2nd: entrectinib	-	5 mos. SD	[62]
NSCLC	C6-R33	NGS	53	F	-	IV	* 1st: crizotinib * 2nd: dabrefenib	Crizotinib resistance: BRAF V600E mutation	2 mos. PR after crizotinib. Dead 15 days after dabrefenib	[136]
Metastatic NSCLC	-	NGS	30	F	-	IIIC (T3N3M0)	1st: cisplatin + pemetrexed 2nd: nedaplatin + pemetrexed * 3rd: crizotinib * 4th: cabozantinib	Crizotinib resistance: MET D1228N mutation	Dead 21 months from dx	[137]
NSCLC- ASC	C6-R34	NGS	43	F	0	IIIA (pT2aN2M0)	1st: albumin-bound paclitaxel + camrelizumab * 2nd: crizotinib 3rd: pemetrexed + carboplatin + bevacizumab 4th: cisplatin + gemcitabine + bevacizumab + * crizotinib	-	Dead ~15 mos. after dx	[138]
NSCLC	C6-R32 C6-R35	DNA NGS RNA NGS FISH Sanger Seq	44	F	0	IVB	* 1st: crizotinib * 2nd: lorlatinib 3rd: chemotherapy	Crizotinib resistance: bone metastasis; lorlatinib resistance: ROS1 G2032R mutation	Dead ~19 mos. after dx	[63]
NSCLC	-	Molecular testing—not specified	54	M	0	IV	* 1st: crizotinib * 2nd: entrectinib 3rd: carboplatin + pemetrexed 4th: pemetrexed + pembrolizumab * 5th: repotrectinib * 6th: cabozantinib	Possible entrectinib resistance: MET amplification	Deceased ~50 months after dx	[139]
NSCLC	-	NGS	54	F	30	IVA (T2aN3M1a)	1st: cisplatin + pemetrexed 2nd: carboplatin + paclitaxel 3rd: pemetrexed 4th: nivolumab 5th: docetaxel 6th: pemetrexed 7th: S-1 8th: gemcitabine * 9th: entrectinib	-	1st: PR 2nd: SD 3rd: PR 4th: SD 5th: PR 6th: PD 7th: PD 8th: PD 9th: PR	[140]
NSCLC	-	NGS	49	F	0	IV	* 1st: entrectinib * 2nd: crizotinib 3rd: carboplatin * 4th: repotrectinib	Possible entrectinib resistance: ROS1 F2004V mutation	1st: PR 2nd: SD 3rd: PD 4th: PR	[141]

RT-PCR: reverse transcription-polymerase chain reaction; NGS: next-generation sequencing; PY: Pack-years; CBPB: classic biphasic pulmonary blastoma; PPC: pulmonary pleomorphic carcinoma; PFS: progression-free survival; CR: complete response; PD: progressive disease; WGS: whole-genome sequencing; SD: stable disease; PR: partial response; ASC: adenosquamous carcinoma; FFPE: formalin-fixed paraffin-embedded; TMA: tissue microarray; TKI: tyrosine kinase inhibitor; BRAF: B-Raf proto-oncogene, serine/threonine kinase gene; dx: diagnosis; Seq: sequencing; CGP: comprehensive genomic profiling; ctDNA: circulating tumor DNA. * indicates TKI prescribed after fusion identification. - indicates data not available. ~ denotes approximately equal to.

**Table 3 ijms-24-15981-t003:** CD74-NRG1 clinical patient characteristics.

Diagnosis	Variant	Detection Method	Age	Gender	Smoker/Pack Year (PY)	Stage	Treatment	Lifespan OS	Reference
NSCLC	-	Genetic Testing	62	F	0	IV	1st: pemetrexed + cisplatin 2nd: atezolizumab + albumin-bound paclitaxel + carboplatin * 3rd: afatinib + pyrotinib	PFS 5 mos.	[143]
IMA	C6-N6	NGS	86	M	0	IIA (pT2bN0M0)	1st: carboplatin + pemetrexed 2nd: paclitaxel + bevacizumab 3rd: nivolumab 4th: GSK2849330 (anti-ERBB3 mAb) * 5th: afatinib	PR 1 year and 7 mos. On GSK2849330	[67]
IMA	C8-N6	Multiplex PCR	81	M	Former, smoked cigars for 1 year	IIB (pT3N0M0)	* 1st: afatinib	SD at 6 wks. and PD at 13 wks.	[67]
IMA	C7-N6	NGS	51	M	<1	IIIB (T4N2M0)	1st: cisplatin + pemetrexed * 2nd: afatinib	PD at 8 wks. after afatinib started	[67]

PFS: progression-free survival, PR: partial response, SD: stable disease, PD: progressive disease. - indicates data not available. * indicates TKI prescribed after fusion identification.

## Data Availability

Not applicable.

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
