# Peer review of "Analysis of CD74 Occurrence in Oncogenic Fusion Proteins"

_ijms, 2023, doi:10.3390/ijms242115981_

Round 1

Reviewer 1 Report

Comments and Suggestions for Authors

This is an analysis of CD74 occurrence in oncogenic fusion proteins. It is a well-organized and well-researched article on CD74-related fusion genes. 

In general, the paper is fine as it is, but it would be even better if the authors added a discussion of the clinical significance of drugs for CD74 and of other genetic variants, such as CD74 amplification, for example.

Comments on the Quality of English Language

OK

Author Response

The authors would like to thank the reviewer for this very useful suggestion. We have added a paragraph to highlight the clinical importance of CD74 therapeutics in malignancies in which CD74 is overexpressed.

Reviewer 2 Report

Comments and Suggestions for Authors

Dear Author,

The review article is well-written in a structured manner, I feel that the data collected, sorted, and presented well will help the readers to understand the CD74 surface receptors.

Thank you

Author Response

We appreciate the time the reviewer took to read through and rate our review. Thank you for looking over the structure, data, and over all presentation. We are glad that this will aid in understanding of the various CD74 receptors and their involvement in oncogenic fusions.

Reviewer 3 Report

Comments and Suggestions for Authors

The paper is indeed an interesting Review that summarizes and put together many findings on the CD74 oncogenic fusion proteins.

The paper is well written and up-to date and in my opinion is suitable for publication

Author Response

  • Thank you for your comments and critique of our paper. We are pleased that the reviewer found the CD74 oncogenic fusion review both current and interesting. Your time in the evaluation of our manuscript is greatly appreciated.